# Impact of employing primary healthcare professionals in emergency department triage on patient flow outcomes: a systematic review and meta-analysis

Maya M Jeyaraman [1], Rachel N Alder,[2] Leslie Copstein,[1] Nameer Al-Yousif,[1] Roger Suss,[3] Ryan Zarychanski,[4] Malcolm B Doupe [5] Simon Berthelot,[6] Jean Mireault,[7] Patrick Tardif,[8] Nicole Askin,[9] Tamara Buchel,[10] Rasheda Rabbani,[1] Thomas Beaudry,[11] Melissa Hartwell,[12] Carolyn Shimmin,[1] Jeanette Edwards,[13] Gayle Halas [14] William Sevcik,[15] Andrea C Tricco [16] Alecs Chochinov,[17] Brian H Rowe,[15,18] Ahmed M Abou-Setta[1]

For numbered affiliations see end of article.

**Correspondence to**
Dr Maya M Jeyaraman;
maya.jeyaraman@umanitoba.ca

## ABSTRACT

**Objectives** To identify, critically appraise and summarise evidence on the impact of employing primary healthcare professionals (PHCPs: family physicians/general practitioners (GPs), nurse practitioners (NP) and nurses with increased authority) in the emergency department (ED) triage, on patient flow outcomes.

**Methods** We searched Medline (Ovid), EMBASE (Ovid), Cochrane Library (Wiley) and CINAHL (EBSCO) (inception to January 2020). Our primary outcome was the time to provider initial assessment (PIA). Secondary outcomes included time to triage, proportion of patients leaving without being seen (LWBS), length of stay (ED LOS), proportion of patients leaving against medical advice (LAMA), number of repeat ED visits and patient satisfaction. Two independent reviewers selected studies, extracted data and assessed study quality using the National Institute for Health and Care Excellence quality assessment tool.

**Results** From 23 973 records, 40 comparative studies including 10 randomised controlled trials (RCTs) and 13 pre–post studies were included. PHCP interventions were led by NP (n=14), GP (n=3) or nurses with increased authority (n=23) at triage. In all studies, PHCP-led intervention effectiveness was compared with the traditional nurse-led triage model. Median duration of the interventions was 6 months. Study quality was generally low (confounding bias); 7 RCTs were classified as moderate quality. Most studies reported that PHCP-led triage interventions decreased the PIA (13/14), ED LOS (29/30), proportion of patients LWBS (8/10), time to triage (3/3) and repeat ED visits (5/6), and increased the patient satisfaction (8/10). The proportion of patients LAMA did not differ between groups (3/3). Evidence from RCTs (n=8) as well as other study designs showed a significant decrease in ED LOS favouring the PHCP-led interventions.

**Conclusions** Overall, PHCP-led triage interventions improved ED patient flow metrics. There was a significant decrease in ED LOS irrespective of the study design, favouring the PHCP-led interventions. Evidence from well-designed high-quality RCTs is required prior to widespread implementation.

## Strengths and limitations of this study

► The main strength of our systematic review is that our study team engaged and collaborated with patient and public partners during the design, conduct and dissemination phases of the study by following the criteria identified for patient-oriented research which emphasises the active and meaningful engagement of patients as research partners.
► This systematic review was conducted using the rigorous Cochrane systematic review methodology and used an a priori registered protocol.
► A main limitation of this systematic review is that we did not include non-English language publications.

**PROSPERO registration number** CRD42020148053.

## INTRODUCTION

Healthcare systems worldwide experience emergency departments (ED) overcrowding,[1–5] which impacts the timely delivery of healthcare,[6 7] patient and provider dissatisfaction,[8] and other adverse outcomes.[9] ED overcrowding is a complex phenomenon and is associated with input (increased patient volume), throughput (ED boarding), and especially output (lack of hospital beds) factors, as well as system-wide influences.[10] A large volume of lower acuity patients presenting to ED leads to demand-capacity mismatch and entry block (eg, delays in ED assessment).[10 11]

Lower acuity ED patients generally include patients: (1) having low acuity triage codes; (2) being discharged quickly or (3) being seen by an alternative primary healthcare provider.[12] These alternative primary

healthcare providers are typically physicians (family physicians/general practitioners (GPs)), nurse practitioners (NP), nurses with increased authority, or physician assistants who are legally authorised to provide or coordinate healthcare to patients.[13] Studies have reported that 8%–62% of all ED presentations are lower acuity.[14–16] With ED visits increasing by 20% each year, along with a decrease in operational EDs,[17] lower acuity visits may lead to unnecessary diagnostic testing, greater healthcare spending, lost opportunity for continuity of care with primary care physicians, suboptimal care due to hurried management and prolonged ED length of stay (LOS).[12 14 18 19] Increased demand for ED services also leads to increased ED wait times and patients choosing to leave ED without being-seen, thus potentially compromising patient safety.[17]

Worldwide, there is growing interest in interventions and strategies, either to discourage lower acuity ED visits or to reduce the impact of lower acuity visits in the ED by improving patient flow. Studies have investigated the impact of interventions such as public and patient education,[14] financial disincentives (higher copayments for lower acuity ED visits),[20] increasing after-hours primary care,[21] patient redirection to non-ED care alternatives[22] and advanced access[23 24] to discourage unnecessary ED utilisation. Since EDs have no control over the volume of presenting patients and ED presentations continue to be on the rise,[14 17] recommendations have been made to focus on strategies to improve patient flow within the ED.[25] Studies have investigated various strategies to improve ED patient flow, including triage related interventions.[8 26]

While the precise role of the NPs, GPs or nurses given increased authority (all referred to as primary healthcare professionals (PHCPs)) in an ED is unclear, they may provide potential benefits to improve ED times and outcomes. Studies have reported the following roles of the PHCPs at ED triage: (1) GP either triaging (seeing and treating, streaming) or supervising triage[27–29]; (2) NP either alone or working alongside a triage nurse (ordering investigations, streaming, seeing and treating, or assessing patients and discharging/redirecting)[17–19 25 30–39]; (3) Triage nurse with increased authority given extra capacities outside of their usual scope of practice to order investigations for patients before streaming to the ED MD.[40–62] Although, many primary research studies have investigated the impact of PHCPs[19 32 36 37 40 63 64] at triage on ED patient flow, to the best of our knowledge, there are no systematic reviews that have summarised evidence from these studies.

The main objective of our systematic review was to identify, critically appraise and summarise evidence on the effectiveness of employing PHCPs at ED triage to improve ED patient flow metrics.

## METHODS
Using an a priori systematic review protocol developed in collaboration with patient partners, we conducted this review according to guidelines enumerated in the Methodological Expectations of Cochrane Intervention Reviews. Our systematic review is reported using the Preferred Reporting Items for Systematic reviews and Meta-Analyses (PRISMA) guideline.[65]

### Eligibility criteria
We included comparative studies (only English language) of any ED triage intervention that involved a PHCP and was designed to improve ED (adult and paediatric) patient flow metrics. We excluded primary studies involving exclusively emergency physicians (ED MD), such as the triage liaison physician (TLP).[26] The primary outcome was the time to provider initial assessment (PIA: time from ED arrival to the time when the patient is first assessed by an ED provider (ED MD, NP or a GP in the ED)). Secondary outcomes were ED LOS (time from ED arrival to disposition), the proportion of patients who left without being seen (LWBS), proportion of patients leaving against medical advice (LAMA), time to triage, number of repeat ED visits and patient satisfaction. The outcome measures were selected a priori in collaboration with the patient partners in the research team. We have reported a more detailed list of the inclusion and exclusion criteria in online supplemental table 1.

### Literature search methods for identifying relevant citations
In conjunction with a health librarian (TR), we designed a search strategy for Medline (Ovid) to identify literature relevant to the objective (from inception to June 2018, and later updated in January 2020). Since most of the potentially relevant studies would be performed in the USA, Europe and Commonwealth countries, search results were limited to English language publications. Our Medline search was peer-reviewed by a second librarian (JJ),[66] principal investigators (MJ and AA-S) and patient partners (MH and TBe). Once finalised, the Medline search strategy (online supplemental table 2) was adapted for replication in the following databases: EMBASE (Ovid), Cochrane Library (Wiley) and CINAHL (EBSCO). An experienced librarian (NA-Y) searched the included databases up to January 2020. The bibliographic search was supplemented with searching the grey literature (ie, difficult to locate unpublished studies) as listed in online supplemental table 3. We also searched the reference lists of all the included publications for additional relevant studies. We used EndNote (V.X7, Thomson Reuters) for reference management.

### Selection of sources of evidence
Two reviewers (RA and (LC or NA-Y)) independently screened the titles and abstracts, and full texts of relevant citations using pilot tested screening forms. Any disagreement on inclusion was resolved through consensus or third party (MJ) adjudication.

### Data extraction, data analysis and quality assessment
Standardised data extraction forms were developed to record data from each of included studies after pilot

testing. At least two review authors independently extracted baseline characteristics (RA, LC and NA-Y), outcome data (RA, LC) and assessed methodological quality (MJ, RA and LC) on these studies. Disagreements among reviewers were resolved through consensus or third-party adjudication (MJ or AA-S). A meta-analysis of mean differences (MD) in ED times with 95% CIs was planned a priori to derive pooled summary estimates. Heterogeneity among included studies was quantified and tested using $I^2$ ($I^2$ statistic and $\chi^2$ statistic, respectively). An $I^2$ value >50% was considered high heterogeneity; we made an a priori methodological decision that heterogeneity indicated by $I^2$ >50% was too high to justify data pooling to generate a summary measure. For studies that did not report any measure of variance we imputed the largest SE from among the included studies. In the event that meta-analysis was not possible, the effect estimates (MD and SE) from included studies reporting data for the primary outcome and ED LOS were depicted in the form of a forest plot for various a priori subgroups (study designs or PHCP interventions). In these cases, where appropriate, the median of the primary study outcome was reported as the average measure.

We assessed the included studies using the National Institute for Health and Care Excellence (NICE) quality appraisal tool for quantitative studies of intervention[67] as it can be used for multiple study designs. A detailed description is reported under online supplemental appendix methods.

### Patient and public involvement

We collaborated with a diverse group of 13 patient partners (self-identified as indigenous, immigrant, white and/or living with disability) during the design phase and the conduct phase of this project, to refine the review question, refine the inclusion criteria, and select patient-important outcomes. Two (TBe and MH) of these patient partners collaborated and supported our grant application to obtain funding for this project. During the conduct phase of this systematic review three patient partners helped refine the search strategy (by identifying missing search terms and suggesting additional search terms in the preliminary search strategy), review to confirm included studies, and in knowledge dissemination (copresented abstract at a conference and co-authoring the manuscript). We have reported the patient partner involvement in this systematic review according to GRIPP2 checklist (short form).[68]

### RESULTS

We identified 23 973 relevant citations from database search, of which 40 met the inclusion criteria[17–19 25 27–62] (44 study reports). The study selection process is reported using the PRISMA study flow chart (figure 1).

### Study characteristics

Included studies were full-length journal articles[17–19 25 27–44 46–48 50 51 55 57 58 60] (n=31; 77.5%), abstracts

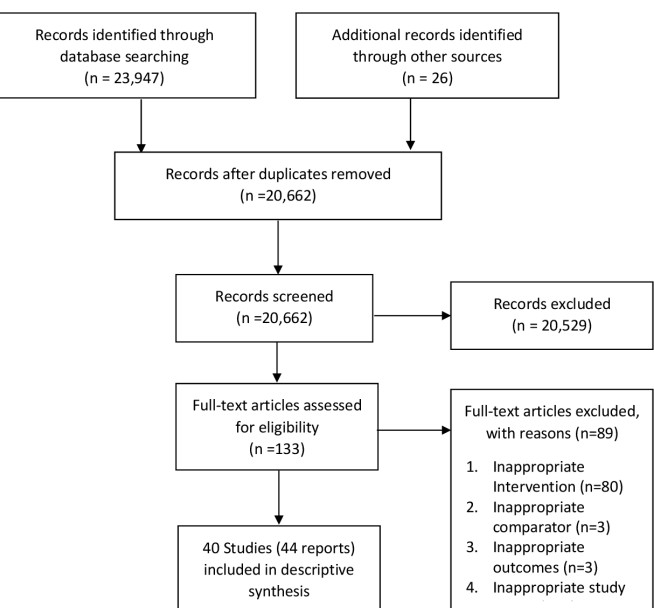

**Figure 1** PRISMA study flow diagram. PRISMA, Preferred Reporting Items for Systematic Reviews and Meta-Analyses.

(n=8)[45 49 52–54 56 59 62] or thesis (n=1)[61] published from 1993 to 2020. More than half were conducted in North America[17 18 25 30 31 33–35 38–41 48 50 51 53 54 57 59 61 62] (n=21; 52.5%), and the rest were conducted in Europe[19 27–29 36 37 42 44 47 52 56] (n=11; 27.5%), Asia[55 58 60] (n=3; 7.5%), Australia[32 43] (n=2; 5%), Middle East[45 46] (n=2; 5%) or the location was not reported[49] (n=1; 2.5%) (table 1). Most studies used a pre–post intervention design[17 18 25 30–34 36–39 61] (n=13; 32.5%) and the remaining were characterised as randomised controlled trials (RCT)[41 42 44 45 50 51 55–57 59] (n=10; 25%), observational retrospective cohort studies[19 28 40 47 53 54 60 62] (n=8, 20%), controlled before and after studies (CBA)[27 35 46 52] (n=4; 10%), quasi-randomised trials[43] (n=1; 2.5%) observational prospective cohort studies[48 49 58] (n=3; 7.5%) or cross-sectional observational studies[29] (n=1; 2.5%). The median duration of intervention reported among included studies was 6 months (range: 2.5 days to 17 months). Studies were mostly conducted in urban EDs[18 19 25 28 30–33 35–38 40–46 48–51 54 57–59 62] (n=28; 70%), one (2.5%) was conducted in a rural ED and two[27 29] (5%) were conducted in a combination of urban and rural facilities; nine[17 34 47 52 53 55 56 60 61] (22.5%) studies did not report their setting. We classified the EDs reported in the included studies into paediatric EDs (age <18)[18 45 50 53 57 60] (n=6; 15%), mixed EDs seeing both adults and children[17 32 48 52 55 58] (n=6; 15%) and adult only EDs[30 33 41 43 46] (n=5; 12.5%), depending on the age of the population that the EDs served. More than half of the studies (n=23, 57.5%) did not specifically report the age of the population that their EDs served.[19 25 27–29 31 34–40 42 44 47 49 51 54 56 59 61 62] The majority (n=15, 38%) of included studies[19 27 28 30 32 33 36–39 43 44 46 55 60] reported enrolling only patients with triage category 4–5 (additional details reported in online supplemental results 1).

**Table 1** Characteristics of included studies

| Study ID (first author, year) | Country; urban/rural ED; adult/paediatric/ mixed ED | Study design | Type of PHCP | Intervention | Duration of intervention (months) | Study quality |
|---|---|---|---|---|---|---|
| Adam *et al*, 2014[45] | Saudi Arabia; urban ED; paediatric ED | RCT | Nurse | Nurse triage-plus | 0.5 | Moderate |
| Al Abri, 2020[46] | Oman; urban ED; adult ED | CBA | Nurse | Nurse triage-plus | NR | Low |
| Al Kadhi, 2017[47] | UK; NR; NR | RC study | Nurse | Nurse triage-plus | 12 | Low |
| Ashurst, 2014[48] | USA; urban ED; mixed ED | PC study | Nurse | Nurse triage-plus | 10 | Low |
| Celona, 2018[30] | USA; urban ED; adult ED | Pre–post | NP | NP Team triage | 12 | Low |
| Cheung, 2002[40] | Canada; urban ED; NR | RC study | Nurse | Nurse triage-plus | NR | Low |
| Day, 2013[31] | USA; urban ED; NR | Pre–post | NP | NP Team triage | 1 | Low |
| Demarco, 2010[49] | NR; urban ED; NR | PC study | Nurse | Nurse triage-plus | 1 | Low |
| Dixon, 2014[50] | Canada; urban ED; paediatric ED | RCT | Nurse | Nurse triage-plus | 12 | Low |
| Edwards, 2011[32] | Australia; urban ED; mixed ED | Pre–post | NP | NP Team triage | 17 | Low |
| Fan, 2006[51] | Canada; urban ED; NR | RCT | Nurse | Nurse triage-plus | 3 | Moderate |
| Fontanel, 2011[52] | France; NR; mixed ED | CBA | Nurse | Nurse triage-plus | 0.08 | Low |
| Gardner, 2018[33] | USA; urban ED; Adult ED | Pre–post | NP | NP Team triage | 0.5 | Low |
| Gaucher, 2010[53] | Canada; NR; paediatric ED | RC study | Nurse | Nurse triage-plus | 12 | Low |
| Hackman, 2015[54] | USA; urban ED; NR | RC study | Nurse | Nurse triage-plus | 17 | Low |
| Hayden, 2014[17] | USA; NR; mixed ED | Pre–post | NP | NP Team triage | 2 | Low |
| Ho, 2018[55] | China; NR; mixed ED | RCT | Nurse | Nurse triage-plus | NR | Moderate |
| Jobé, 2019[56] | France; NR; NR | RCT | Nurse | Nurse triage-plus | NR | Low |
| Klassen, 1993[57] | Canada; urban ED; paediatric ED | RCT | Nurse | Nurse triage-plus | 12 | Low |
| Kool, 2008[27] | Netherlands; both; NR | CBA | GP | GP team triage | 12 | Low |
| Lee, 1996[58] | China; urban ED; mixed ED | PC study | Nurse | Nurse triage-plus | 3 | Low |
| Lee, 2014[59] | Canada, urban ED; NR | RCT | Nurse | Nurse triage-plus | 12 | Moderate |
| Lee, 2016[41] | Canada; urban ED; adult ED | RCT | Nurse | Nurse triage-plus | 12 | Moderate |
| Li, 2018[60] | China; NR; paediatric ED | RC study | Nurse | Nurse triage-plus | 5 | Low |
| Lijuan, 2017[61] | USA; NR; NR | Pre–post | Nurse | Nurse triage-plus | 6 | Low |
| Lindley-Jones, 2000[42] | England; urban ED; NR | RCT | Nurse | Nurse triage-plus | 12 | Moderate |
| Love, 2012[25] | USA; urban ED; NR | Pre–post | NP | NP Team triage | 0.5 | Low |
| MacKenzie, 2015[34] | USA; NR; NR | Pre–post | NP | NP Team triage | 2 | Low |
| Parris, 1997[43] | Australia; urban ED; adult ED | Quasi-RCT | Nurse | Nurse triage-plus | 6 | Low |
| Pierce, 2016[35] | USA; urban ED; NR | CBA | NP | NP Team triage | 5.5 | Low |
| Rogers, 2004[36] | England; urban ED; NR | Pre–post | NP | NP Team triage | 12 | Low |
| Shrimpling, 2002[37] | England; urban ED; NR | Pre–post | NP | NP Team triage | 0.75 | Low |
| Sikkenga, 2016[62] | USA; urban ED; NR | RC study | Nurse | Nurse triage-plus | NR | Low |
| Thurston, 1996[44] | England; urban ED; NR | RCT | Nurse | Nurse triage-plus | 2 | Moderate |
| Tsai, 2012[18] | USA; urban ED; paediatric ED | Pre–post | NP | NP Team triage | NR | Low |

Continued

**Table 1** Continued

| Study ID (first author, year) | Country; urban/rural ED; adult/paediatric/mixed ED | Study design | Type of PHCP | Intervention | Duration of intervention (months) | Study quality |
|---|---|---|---|---|---|---|
| Tucker, 2015[38] | USA; urban ED; NR | Pre–post | NP | NP Team triage | 6 | Low |
| Uthman, 2018[19] | England; urban ED; NR | RC study | NP | NP Team triage | 12 | Low |
| van den Bersselaar *et al*, 2018[28] | Netherlands; urban ED; NR | RC study | GP | GP team triage | 11 | Low |
| van Gils-van Rooij, 2018[29] | Netherlands; both; NR | CS study | GP | GP team triage | NA | Low |
| Zager, 2018[39] | USA; rural ED; NR | Pre–post | NP | NP Team triage | 4 | Low |

CBA, controlled before and after; CS study, cross-sectional observational study; ED, emergency department; GP, general practitioner; NA, not applicable; NP, nurse practitioner; NR, not reported; PHCP, primary healthcare provider; RC study, retrospective cohort study; RCT, randomised controlled trial.

The majority (82.5%) of included studies were of low methodological quality and the remaining seven (17.5%) included studies (RCTs)[41 42 44 45 51 55 59] were of a moderate methodological quality (table 1 and online supplemental table 4).

We categorised the triage interventions involving PHCP reported by the included studies, in comparison to the traditional (nurse-led) triage model (figure 2), as follows: (1) GP team-triage[27–29] (n=3, 7%): where GP was involved in the ED triage (triaging or supervising triage) either seeing and treating low-acuity patients or streaming moderate to high-acuity patients to the ED MD; (2) NP team-triage[17–19 25 30–39] (n=14, 35%): where the NP was located at the ED triage area working alongside a triage nurse, either ordering investigations at triage before streaming to ED MD, seeing and treating low-acuity patients, directing low-acuity patients to a GP located within ED for treatment, or assessing patients and discharging/redirecting with a same day appointment with a GP at an adjoining GP clinic; (3) Nurse triage-plus[40–62] (n=23, 58%): triage nurse with increased authority (extra capacities outside of their usual scope of practice) to order investigations for patients before

streaming to the ED MD. The traditional ED care model with an ED nurse-led triage, followed by the ED MD assessment was considered standard of care and the comparator in all the included studies (figure 2).

### Provider initial assessment

Fourteen studies[17 18 25 27 29 30 33–36 38 42 46 55] (35%) reported the effect of PHCP triage interventions on PIA in comparison to a traditional nurse-led triage. Using a forest plot, we depicted the effectiveness of the PHCP triage interventions on PIA subgrouped by study design (figure 3). Two RCTs[42 55] (of moderate quality), reported a non-significant small decrease in PIA in the PHCP triage intervention (nurse triage-plus) group compared with the traditional nurse-led triage model (mean difference (MD) −0.36 min (95% CI −4.53 to 3.81); two studies; I²: 39%; p=0.20; moderate quality). Three CBA studies[27 35 46] (low quality) reported a decrease in PIA in the PHCP

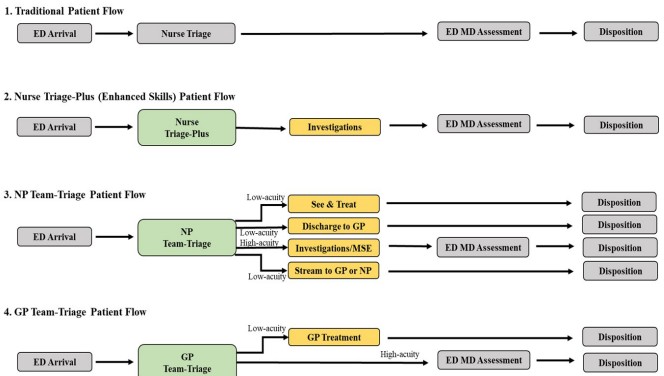

**Figure 2** Various models for PHCP involvement in triage of emergency department patients. ED, emergency department; GP, general practitioner; MD, mean difference; NP, nurse practitioner; PHCP, primary healthcare professional.

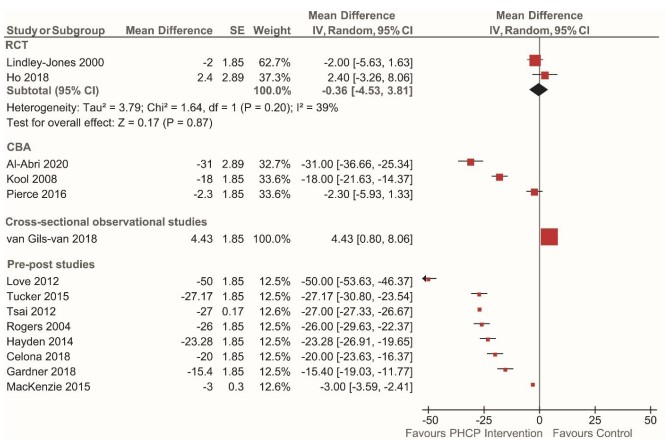

**Figure 3** Effectiveness of primary healthcare professional (PHCP) interventions on time to provider initial assessment (in minutes) subgrouped by study design. The horizontal black lines represent 95% CIs and the red dots in the middle represents point estimates (mean difference). CBA, controlled before and after; RCT, randomised controlled trial.

triage intervention group (median (range) = −18 min (95% CI −2.3 to −31)).

All eight pre–post studies[17 18 25 30 33 34 36 38] (low quality) reported a significant decrease in PIA (median (range)=−24.65 min (95% CI −3 to −50)) in the PHCP triage intervention group, compared with the traditional nurse-led triage model. Exploration of heterogeneity among pre–post studies ($I^2$: 100%) revealed four studies[25 30 33 34] that contributed to all the observed heterogeneity; however, we were unable to identify specific reasons for heterogeneity. A sensitivity analysis without these four studies showed a significant mean decrease of PIA by 26 min favouring the PHCP triage intervention group (NP team triage). One cross-sectional observational study[29] (low quality) failed to identify a difference in PIA in the PHCP intervention group (4.4 min). The results for PIA sub-grouped by various PHCP interventions is reported under online supplemental appendix results 2. We have depicted the effectiveness of each of the three models of PHCP triage interventions on PIA separately using a forest plot (online supplemental figure 1).

Three studies[25 32 36 37] reported greater percentage of patients seen within benchmark times in the NP team triage intervention groups compared with the traditional nurse-led triage model (online supplemental figure 2). A fourth study[32] reported that greater percentage of patients (all ATS categories) were seen within benchmark times in the NP team triage group compared with the traditional nurse-led triage group (data not shown).

## ED LOS

ED LOS was reported by thirty studies (75%).[17 18 29 31 33–36 38–41 43–52 54–60 62] Using a forest plot, we have depicted the effectiveness of the PHCP triage interventions on ED LOS sub-grouped by study design (figure 4). Eight RCTs[41 44 45 50 51 55 57 59] (six[42 45 46 52 56 60] of moderate quality and two[50 57] of low quality), reported a significant decrease in ED LOS (MD −15.31 min (95% CI −18.35 to −12.27); eight studies; $I^2$: 0%; p<0.00001) in the PHCP triage intervention (nurse triage-plus) group compared with the traditional nurse-led triage model. The CBA studies[35 46 52] (low quality) reported a significant decrease in ED LOS (mean difference −63.17 min (95% CI −101.93 to −24.40); three studies; $I^2$: 51%; p=0.001) in the PHCP triage intervention group (two nurse-triage plus and one NP team triage) compared with the traditional nurse-led triage model and the three retrospective cohorts[40 54 60] (low quality) also reported a significant decrease in the ED LOS (MD −13.96 min (95% CI −19.31 to −8.61); three studies; $I^2$: 37%; p<0.00001) in the PHCP triage intervention group (nurse triage-plus), compared with the traditional nurse-led triage model.

Among eight pre–post studies[17 18 31 33 34 36 38 39] (low quality), all reported a decrease (five were significant) in ED LOS (median (range) = −28 min (95% CI −16.65 to −102) favouring the PHCP triage intervention group (NP team triage). Exploration of heterogeneity among pre–post studies ($I^2$: 99%) revealed four studies[18 31 33 39] that

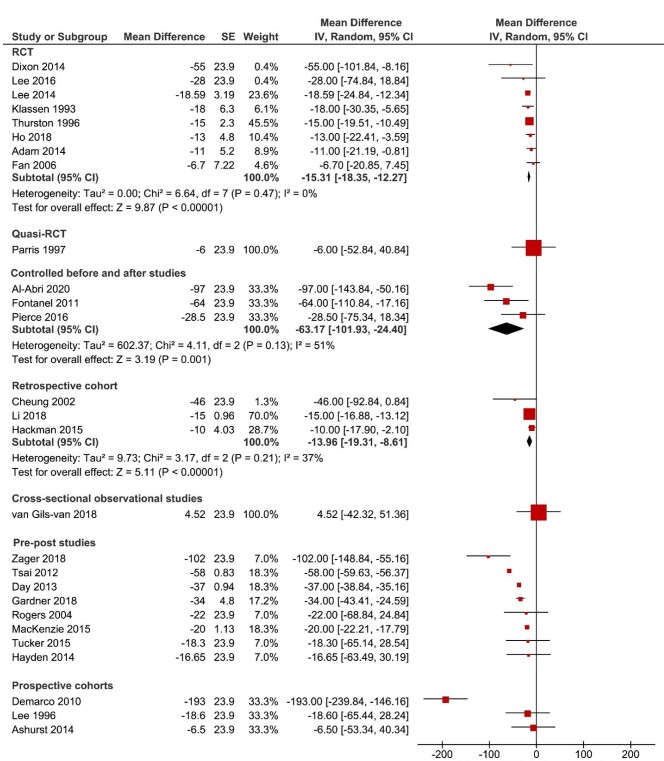

**Figure 4** Effectiveness of primary healthcare professional (PHCP) interventions on ED Los (in minutes) subgrouped by study design. The horizontal black lines represent 95% CIs and the red dots in the middle represents point estimates (mean difference). ED, emergency department; RCT, randomised controlled trial.

contributed to all the observed heterogeneity; however, we were unable to identify specific reasons for heterogeneity. A sensitivity analysis without these four studies showed a significant mean decrease of ED LOS by 17 min favouring the PHCP triage intervention group (NP team triage). One quasi-RCT,[43] and one cross-sectional observational study[29] reported no significant differences in ED LOS between comparison groups. Among the three prospective observational cohorts,[48 49 58] one reported significant decrease in ED LOS whereas other two reported a non-significant decrease in ED LOS favouring PHCP intervention group. The ED LOS subgrouped by various PHCP interventions is reported under online supplemental appendix results 3. We have depicted the effectiveness of each of the three models of PHCP triage interventions on ED LOS separately, using a forest plot (online supplemental figure 3).

## Other outcomes

Ten studies[17–19 25 30 33 34 38 60 61] reported data for percentage of patients LWBS (table 2). Eight studies reported a reduction in percentage of patient LWBS in the NP team triage intervention group, except one[17] (five[18 19 25 33 34] reported statistically significant decrease). Two[60 61] studies reported a non-significant decrease in percentage of patients LWBS in nurse triage-plus intervention group. The median effect of all estimates is a reduction in LWBS of −2.31% (IQR:

**Table 2** Leave without being seen outcome data reported by included studies

| Study ID (first author, year) | Triage intervention | Study design | Intervention (%) | Comparator (%) | Percentage difference | Reported statistical significance |
|---|---|---|---|---|---|---|
| Celona, 2018[30] | NP team triage | Pre–post | 4.7 | 3.3 | 1.4 | NR |
| Love, 2012[25] | NP team triage | Pre–post | 0.93 | 3.39 | –2.46 | Significant |
| MacKenzie 2015[34] | NP team triage | Pre–post | 0.7333 | 2.96 | –2.2267 | Significant |
| Gardner, 2017[33] | NP team triage | Pre–post | 2.2 | 4.6 | –2.4 | Significant |
| Hayden, 2014[17] | NP team triage | Pre–post | 5.8 | 5.4 | 0.4 | NS |
| Tsai, 2012[18] | NP team triage | Pre–post | 3 | 9.7 | –6.7 | Significant |
| Tucker, 2015[38] | NP team triage | Pre–post | 1.3 | 5.07 | –3.77 | NR |
| Uthman, 2018[19] | NP team triage | Retrospective cohort | 2.2 | 3.9 | –1.7 | Significant |
| Li, 2018[60] | Nurse triage-plus | Retrospective cohort | 0.7 | 6.9 | –6.2 | NS |
| Lijuan, 2017[61] | Nurse triage-plus | Pre–post | 7.13 | 7.52 | –0.39 | NS |

NP, nurse practitioner; NR, not reported; NS, not significant.

–0.39 to –3.77). Three pre–post studies[17 34 38] reported the effect of NP team triage intervention on percentage of patients discharged as LAMA (table 3). One study showed a non-significant decrease favouring the intervention group and another two showed a non-significant increase in the percentage of patients LAMA.

Six studies[27 28 31 38 39 53] reported the impact of PHCP interventions on the number of repeat ED visits, and the majority of them reported a decrease in the number of repeat ED visits after PHCP intervention (online supplemental appendix results 4). Ten studies[17 27 33 38 41 48 49 51 59 61] reported the effect of PHCP intervention on patient satisfaction, and the majority of them reported an increase in patient satisfaction (online supplemental appendix results 5). Three studies[34 36 55] reported the impact of PHCP interventions on the time to triage, and all of them reported a decrease in the time to triage after PHCP intervention (online supplemental appendix results 6).

## DISCUSSION
### Main findings
This systematic review has summarised the best available evidence from 40 unique comparative studies on the effectiveness of the PHCP triage interventions to improve ED patient flow metrics and mitigate the negative impacts of ED overcrowding. The findings in this systematic review shows that the PHCP-led triage interventions significantly decrease the ED LOS and lead to improvements in key ED patient flow metrics such as PIA, proportion of patients who LWBS, triage time, ED visits and patient satisfaction.

Although this systematic review highlights the positive impact of three unique PHCP triage models on key ED patient flow metrics, it is important to note that the most comprehensive evidence (data for the primary review outcome and all of the secondary outcomes) was available mainly for the nurse triage-plus and NP team triage models, with the least evidence available for the GP team triage model.

### Comparison with other reviews
To the best of our knowledge, this is the first review to investigate specific triage interventions involving NPs and GPs. Previous work had focused specifically on the impact of TLP[26] or triage nurse ordering[69] on ED patient flow metrics. Rowe et al[26] investigated the impact of TLP's and reported reductions in ED LOS and PIA. However, the interventions mainly involved emergency physicians.

**Table 3** Leave against medical advice outcome data reported by included studies

| Study ID (first author, year) | Triage intervention | Study design | Intervention (%) | Comparator (%) | Percentage difference | Reported statistical significance |
|---|---|---|---|---|---|---|
| MacKenzie, 2015[34] | NP team triage | Pre–post | 0.22 | 0.33 | –0.11 | NS |
| Tucker, 2015[38] | NP team triage | Pre–post | 1.41 | 1.29 | 0.12 | NS |
| Hayden, 2014[17] | NP team triage | Pre–post | 1.4 | 0.06 | 1.34 | NR |

NP, nurse practitioner; NR, not reported; NS, not significant.

Previously, Jennings *et al*[70] published a systematic review on the impact of emergency NP services in the ED and narratively concluded that although not enough data were available for meta-analysis, NPs within ED may have a positive impact on waiting times, patient satisfaction, and quality of care. Again, this review did not focus on NP at triage. A recent Cochrane review investigated the role of primary care professionals (emergency NP and GP) in the ED[71] and concluded that due to limited evidence and suspected bias in allocations of ED patients it was unclear if hiring primary care professionals would decrease PIA, ED LOS and other ED metrics. It is important to note, however, that this Cochrane review did not investigate the role of primary care professionals at ED triage.

## ED LOS and PIA

ED wait times for care delivery is a key performance indicator in many ED settings and our systematic review findings indicate that the PHCP-led triage intervention consistently decreases ED wait times (PIA) and ED LOS. In this review, pre–post studies contributed to the majority of the evidence for effectiveness of PIA (NP team triage). Although heterogeneous and of low quality, the results indicate important potential for the role of NPs in the triage process to reduce ED wait times, improve patient satisfaction and other key ED metrics. A significant decrease in ED LOS was observed with the RCTs (median: −16.8 min) although this was comparatively smaller than the significant decrease observed with the CBA (median: −64 min) or the pre–post studies (median: −28 min). As the minimal clinically important difference in ED LOS is generally accepted to be approximately 30 min (clinically significant), the PHCP-led triage interventions could potentially have a positive impact on ED LOS, if implemented.

## Type of PHCP

One may argue that similar results could be seen with ED MD at triage. Although true, the cost of adding an NP could be far less than adding an ED MD.[38] In our review, we found only three studies reporting evidence on the role of GP's in ED triage, with one CBA study[27] reporting statistically significant decrease in PIA, and a cross-sectional study[29] reporting an increase in PIA when triaged and treated by a GP. The increase in PIA, however, was reported to be due to an increase in the number of self-referrals in order to be seen by the GP involved in triage and treatment of low-acuity patients.[29] In the reported GP team triage interventions, an ED and a GP clinic were co-located and had a joint common entrance, with the GP assistant (supervised by GP) and/or a GP being responsible for the triage of patients (for both ED and GP clinic) and for the treatment of low-acuity patients. The third study[28] reported that GP team triage and x-ray requests at the joint triage reduced the annual ED patient visits. High-quality studies investigating the effectiveness of GPs at ED triage would be valuable.

In our review, the evidence on effectiveness of nurse triage-plus model came mostly from moderate quality studies (RCT or CBA) and showed significant decrease in PIA, ED LOS, and an improved patient satisfaction. Many factors such as patient acuity, EMS traffic/volume and referral patterns often dictate the degree of ED crowding and each ED has their own 'signature'. For example, in settings where most patients present with ambulatory, single system problems, a nurse triage with extra skills might be effective. Conversely, a nurse triage-plus intervention may be less effective when faced with the challenges of an ED setting with high volumes of trauma, EMS traffic and high acuity patients.

It would be generally expected that the addition of any qualified staff in the ED, including addition of NP in triage, would tend to make efficiency of the ED operations better. Although we did not assess staff satisfaction in our review, it would be intuitive to think that the addition of NP in the ED triage may also help improve ED staff satisfaction. Many government-funded EDs are cash-constrained and often cannot add additional resource without strong justification and/or reducing funding elsewhere. While addition of NPs, TLPs or GPs at triage may help, there is still lack of published comparative effectiveness and economic evaluation research to produce a clear cost-effectiveness recommendation. While comparative effectiveness research may prove logistically difficult in the ED and outcome measurements need to be granular and robust (eg, including intended and unintended consequences), these studies are critical to developing recommendations.

Overall, the evidence synthesised by our review indicates that the PHCP-led triage interventions significantly decrease PIA or ED LOS compared with the traditional nurse-led triage model. The studies in this review demonstrate promise to improve ED patient flow metrics by either seeing and treating non-urgent patients in the triage area, starting investigations at triage for moderate to low acuity patients, or assessing and making decision to re-direct very low-acuity patients to an adjoining GP clinic with same day appointments. All of these could mitigate ED overcrowding. Since ED wait times are multi-factorial it cannot be expected that one solution will solve such a complex problem. Each ED will need an individualised approach. Moreover, while calling for improved research quality, we believe comparative effectiveness studies with health economic outcomes are required to fully weigh the costs and benefits associated with any intervention.

## Strengths and limitations

We acknowledge the following limitations in interpreting the results of this systematic review. All systematic reviews are susceptible to publication and selection bias. Selection bias was minimised by using a comprehensive, peer-reviewed search strategy developed by an experienced information specialist. Selection bias was also addressed by using two independent reviewers and third-party adjudication. We evaluated the quality of each included study

using the NICE Quality Appraisal Tool[67]; that is tailored to quantitative studies investigating public health interventions. A few included studies reported the effectiveness of GP team triage intervention on the review outcomes, thus limiting conclusions on GP-led triage interventions. Most of the included studies were of pre–post intervention design providing low quality evidence. Even the included RCTs were only of moderate quality, thus evidence from high-quality studies is lacking, limiting the confidence that can be placed on the results. Nevertheless, irrespective of the study design, we observed a significant decrease in ED LOS favouring PHCP-led triage intervention. Although we used a comprehensive search strategy, for the sake of feasibility we did not consider non-English language studies and the possibility of missing some of the other language studies remains. Despite the compressive search strategy, publication bias is likely since many operational studies never reach publication and many of those would be negative. We also encountered issues with missing data in some of the included studies and resorted to imputation techniques as we were unable to obtain data from study authors. The included studies in this systematic review did not focus on clinical outcomes, such as delayed or missed diagnosis, but it would be important for future studies to quantify relevant clinical outcomes. As included studies were conducted in various countries, health systems and societal contexts, the results from one may not be compatible with evidence from other jurisdictions.

Notwithstanding the above concerns, we believe this review has many strengths, including the rigorous Cochrane systematic review methodology employed and the use of an a priori registered protocol. In addition, our study team included patient partners who collaborated with the investigators during the design, conduct and dissemination phases of the study. Following the criteria identified for patient-oriented research which emphasises the active and meaningful engagement of patients as research partners, twelve diverse group of patient partners from three Canadian provinces (Manitoba, Alberta and Quebec) were engaged from the design stage and throughout the research process around decisions and in knowledge dissemination.

## CONCLUSIONS

PHCP-led triage interventions could be an effective strategy to improve ED patient flow overall by decreasing ED LOS, PIA, time to triage or ED visits, and by improving patient satisfaction. While these triage interventions may work in specific settings, each ED is unique, and policy would have to be evaluated specific to that facility and system. High quality methods are also necessary to further support PHCPs role in ED triage, and it is important for future studies to focus on cost efficiency or incremental value for money as these are critical real-world issues. Additionally, future research could focus on generating high quality evidence on the effectiveness of GP triage

intervention. The acceptability of a PHCP-led interventions in an ED could also be formally ascertained in future studies as experience and beliefs of ED staff may play a role in the success or failure of the policy to implement PHCPs in triage. Finally, the research gap involving rural EDs needs to be addressed.

**Author affiliations**
[1]George and Fay Yee Center for Healthcare Innovation, University of Manitoba, Winnipeg, Manitoba, Canada
[2]Max Rady College of Medicine, Rady Faculty of Health Sciences, University of Manitoba, Winnipeg, Manitoba, Canada
[3]Department of Family Medicine, Rady Faculty of Health Sciences, University of Manitoba, Winnipeg, Manitoba, Canada
[4]Department of Medical Oncology and Hematology, Cancer Care Manitoba, Winnipeg, Manitoba, Canada
[5]Department of Community Health Sciences, University of Manitoba, Winnipeg, Manitoba, Canada
[6]Centre de recherche du CHU de Québec-Université Laval, Axe Santé des populations et Pratiques optimales en santé, Laval, Quebec, Canada
[7]HEC Pôle santé, Université de Montréal, Montreal, Quebec, Canada
[8]Department of Emergency Medicine, Cité de la santé de Laval, Laval, Quebec, Canada
[9]WRHA Virtual Library, University of Manitoba, Winnipeg, Manitoba, Canada
[10]Manitoba College of Family Physicians, Winnipeg, Manitoba, Canada
[11]Patient and Public Engagement Collaborative Partnership, George & Fay Yee Center for Healthcare Innovation, Winnipeg, Manitoba, Canada
[12]Primary and Integrated Health care Innovation Network, Edmonton, Alberta, Canada
[13]Community Health Quality and Learning, Shared Health Manitoba, Winnipeg, Manitoba, Canada
[14]Manitoba Primary and Integrated Health care Innovation Network, Winnipeg, Manitoba, Canada
[15]Department of Emergency Medicine, Faculty of Medicine & Dentistry, University of Alberta, Edmonton, Alberta, Canada
[16]Knowledge Translation Program, St. Michael's Hospital Li Ka Shing Knowledge Institute, Unity Health Toronto, Toronto, Ontario, Canada
[17]Department of Emergency Medicine, Faculty of Medicine, University of Manitoba, Winnipeg, Manitoba, Canada
[18]School of Public Health, University of Alberta, Edmonton, Alberta, Canada

**Acknowledgements** We are very grateful for the Canadian Institutes of Health Research (CIHR) grant (NKS 158643) and the Manitoba Medical Services Foundation (#8-2018-06) and Winnipeg Foundation (#8-2018-06) for providing financial support for this project. Dr. Rowe's research was supported by a Scientific Director's Grant (SOP 168483) from CIHR. Dr. Tricco's research is funded by a Tier 2 Canada Research Chair in Knowledge Synthesis. We are also very grateful to all the patient partners who collaborated with us throughout the duration of this project. We thank Dr. Cristina Villa-Roel for her support during grant and manuscript preparation and the Emergency Medicine Research Group (EMeRG) in the Department of Emergency Medicine at the University of Alberta for in-kind resources. We are very grateful to all the patient partners who collaborated with us in the design, conduct and dissemination stages of this project. We are thankful to Frank Krupka for the kind support of this project during his time as the Executive Director of CHI. We thank Tamara Rader, Christine Nielson, and Janet Joyce for help with the search strategy for this study. We thank our partner organisations (Strategy for Patient-Oriented Research Support for People and Patient-Oriented Research and Trials units (Manitoba, Alberta and Quebec), EMeRG, and Network in Primary and Integrated Healthcare Innovations), and the knowledge users (Winnipeg Regional Health Authority, Emergency Strategic Clinical Network, Shared Health Manitoba, and Manitoba College of Family Physicians) for their kind support and feedback throughout the duration of this project.

**Contributors** MMJ, AMA-S, TBe and MH contributed to the design and conception of the study. MMJ drafted the manuscript with feedback from coauthors. MMJ, AMA-S and RR contributed to the data analysis. NA contributed to developing the search strategy. LC, RNA and NA-Y were involved in study selection process and in data extraction. MBD, SB, JM, PT, BHR, AC, WS brought content expertise in emergency medicine. RS, GH, TBu and JE brought content expertise in family

medicine. CS brought content expertise in patient engagement. ACT, RZ, MMJ and AMA-S brought content expertise in systematic review methodology. All study authors approved the final manuscript. MMJ is responsible for the overall content as guarantor. The guarantor accepts full responsibility for the finished work and/or the conduct of the study, had access to the data, and controlled the decision to publish.

**Funding** This work was supported by research grant from Canadian Institutes of Health Research (NKS 158643), Manitoba Medical Services Foundation (# 8-2018-06) and Winnipeg Foundation (# 8-2018-06).

**Competing interests** None declared.

**Patient and public involvement** Patients and/or the public were involved in the design, or conduct, or reporting, or dissemination plans of this research. Refer to the Methods section for further details.

**Patient consent for publication** Not applicable.

**Provenance and peer review** Not commissioned; externally peer reviewed.

**Data availability statement** All data relevant to the study are included in the article or uploaded as online supplemental information.

**ORCID iDs**
Maya M Jeyaraman http://orcid.org/0000-0002-1548-3987
Malcolm B Doupe http://orcid.org/0000-0002-6889-9097
Gayle Halas http://orcid.org/0000-0003-0433-0632
Andrea C Tricco http://orcid.org/0000-0002-4114-8971

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
