## [Reviewer comments · BMJ Open]

ARTICLE DETAILS

TITLE (PROVISIONAL)	The impact of employing primary healthcare professionals in emergency department triage on patient flow outcomes: A systematic review and meta-analysis
AUTHORS	Jeyaraman, Maya; Alder, Rachel; Copstein, Leslie; Al-Yousif, Nameer; Suss, Roger; Zarychanski, Ryan; Doupe, Malcolm; Berthelot, Simon; Mireault, Jean; Tardif, Patrick; Askin, Nicole; Buchel, Tamara; Rabbani, Rasheda; Beaudry, Thomas; Hartwell, Melissa; Shimmin, Carolyn; Edwards, Jeanette; Halas, Gayle; Sevcik, William; Tricco, Andrea; Chochinov, Alecs; Rowe, Brian; Abou-Setta, Ahmed

VERSION 1 – REVIEW

REVIEWER	Gilbert, Allison University Hospital Center Liege, Emergency Medicine
REVIEW RETURNED	30-Jul-2021

GENERAL COMMENTS	Brief summary and positive aspects: Orso, Daniele University of Udine This systematic review and meta-analysis, the authors approach the relevant topic of Emergency Department overcrowding management with specific triage strategies, in particular primary healthcare providers-led triage, and discuss the outcomes in terms of patient's flow improvements. The introduction clearly evokes the issues that emergency physicians have to face every day. The collaboration with patient partners to create the a priori systematic review protocol is an interesting way to conduct the study. As regards the methodological points, the PRISMA checklist is adequately followed, the PICO is detailed and the PRISMA study flow chart is presented as an appendix. The search was appropriately made in more than 3 different databases (Medline, Embase, Cochrane Library, Cinahl) and the grey literature was considered. The quality of the selected studies was also evaluated using a validated tool. The presence of a meta-analysis to investigate ED times adds a significant interest to the article. As regards the conclusion, few studies on GP-led triage are available and reduced the value of the conclusions made on their efficiency compared to other providers-led triages. These biases are mentioned in the appropriate section of the article. Comments:
--

	All of the aforementioned points highlight the relevance and interest of this article. However, in my opinion, some suggestions can be made:  1. In the introduction section, the authors evoked different healthcare functions: nurse practitioners, nurses with increased authorities, general practitioners and physician assistants. However, no clear definition and/or clear statement about the difference is given at this stage of the article. It could be confusing for the reader who is not familiar with these particular functions. Indeed, these roles are not represented in all countries around the world. The functions should be better described while this is an important point to understand the difference in outcomes. However, we could find a quick description developed in the results section. As the article focuses on the different organization at ED triage involving primary healthcare providers, it could be of interest to add some general clarifications about the role, qualification and ability of the different providers (traditional ED nurse, nurse practitioner, nurse with increased authorities and GP at ED triage). 2. The multiple abbreviations used in the abstract make it somehow difficult to clearly understand. 3. In the discussion section, please pay attention to keep a clear structure of the text and maybe sub-sections could be helpful (e.g. summary of evidence or main findings, strength and limitations, etc).
--	---

REVIEWER	Orso, Daniele University of Udine
REVIEW RETURNED	30-Sep-2021

GENERAL COMMENTS	I congratulate the authors for the systematic review they have proposed. It is a very perceived topic within the organization of the Emergency Departments, and the conclusions of the study are potentially very impactful. The systematic search for the studies to be included is adequate, complete, and well described. The aims are well defined. The statistics are adequate, and the results are well exposed clearly and comprehensively. The discussion is complete and not lengthy. The only aspect that I would clarify is the quantification of "cases of missed diagnosis" in the PHCP-led triage: in other words - although the included studies do not mention it - it would be useful to know how many cases (in percentage) of "missed" diagnoses are verified. It would be useful to add this aspect in the Discussion section (in the Limitations) to orient any further studies in the field adequately. In fact, in addition to the problem of "time spent in ED", I believe the adequacy of the PHCP intervention at the time of triage is important.
---

VERSION 1 – AUTHOR RESPONSE

Reviewer 1

Dr. Allison Gilbert, University Hospital Center Liege

Comments to the Author:

Brief summary and positive aspects:

Through this systematic review and meta-analysis, the authors approach the relevant topic of Emergency Department overcrowding management with specific triage strategies, in particular primary healthcare providers-led triage, and discuss the outcomes in terms of patient's flow improvements. The introduction clearly evokes the issues that emergency physicians have to face every day. The collaboration with patient partners to create the a priori systematic review protocol is an interesting way to conduct the study. As regards the methodological points, the PRISMA checklist is adequately followed, the PICO is detailed and the PRISMA study flow chart is presented as an appendix. The search was appropriately made in more than 3 different databases (Medline, Embase, Cochrane Library, Cinahl) and the grey literature was considered. The quality of the selected studies was also evaluated using a validated tool. The presence of a meta-analysis to investigate ED times adds a significant interest to the article. As regards the conclusion, few studies on GP-led triage are available and reduced the value of the conclusions made on their efficiency compared to other providers-led triages. These biases are mentioned in the appropriate section of the article.

Author response: Thank you kindly for your feedback. Much appreciated.

Comments:

All of the aforementioned points highlight the relevance and interest of this article. However, in my opinion, some suggestions can be made:

1. In the introduction section, the authors evoked different healthcare functions: nurse practitioners, nurses with increased authorities, general practitioners and physician assistants. However, no clear definition and/or clear statement about the difference is given at this stage of the article. It could be confusing for the reader who is not familiar with these particular functions. Indeed, these roles are not represented in all countries around the world. The functions should be better described while this is an important point to understand the difference in outcomes. However, we could find a quick description developed in the results section. As the article focuses on the different organization at ED triage involving primary healthcare providers, it could be of interest to add some general clarifications about the role, qualification and ability of the different providers (traditional ED nurse, nurse practitioner, nurse with increased authorities and GP at ED triage).

Author response: Thank you. In page 13 of the manuscript (results section), we had provided a description of the roles of the various primary healthcare providers. As suggested, we have added a brief description/general clarification about the roles of different providers in the introduction section. "Studies have reported the following roles of the PHCPs at ED triage:: (1) GP either triaging (seeing and treating, streaming) or supervising triage; (2) NP either alone or working alongside a triage nurse (ordering investigations, streaming, seeing and treating, or assessing patients and discharging/re-directing); (3) Triage nurse with increased authority given extra capacities outside of their usual scope of practice to order investigations for patients before streaming to the ED MD."

2. The multiple abbreviations used in the abstract make it somehow difficult to clearly understand.

Author response: Thank you. We have moved the list of abbreviations from page 25 to page 3 (after the title page) of the manuscript to help guide the readers.

3. In the discussion section, please pay attention to keep a clear structure of the text and maybe sub-sections could be helpful (e.g. summary of evidence or main findings, strength and limitations, etc).

Author response: Thank you. As suggested, we have included subheadings to the discussion section.

Reviewer: 2

Dr. Daniele Orso, University of Udine

Comments to the Author:

I congratulate the authors for the systematic review they have proposed. It is a very perceived topic within the organization of the Emergency Departments, and the conclusions of the study are potentially very impactful. The systematic search for the studies to be included is adequate, complete, and well described. The aims are well defined. The statistics are adequate, and the results are well exposed clearly and comprehensively. The discussion is complete and not lengthy.

Author response: Thank you kindly for your feedback. Much appreciated.

The only aspect that I would clarify is the quantification of "cases of missed diagnosis" in the PHCP-led triage: in other words - although the included studies do not mention it - it would be useful to know how many cases (in percentage) of "missed" diagnoses are verified. It would be useful to add this aspect in the Discussion section (in the Limitations) to orient any further studies in the field adequately. In fact, in addition to the problem of "time spent in ED", I believe the adequacy of the PHCP intervention at the time of triage is important.

Author response: Thank you. We agree. As you suggest, we have added a statement in the limitations section regarding this.

"The included studied in this systematic review did not focus on clinical outcomes, such as delayed or missed diagnosis, but it would be important for future studies to quantify relevant clinical outcomes"

VERSION 2 – REVIEW

REVIEWER	Gilbert, Allison University Hospital Center Liege, Emergency Medicine
REVIEW RETURNED	10-Feb-2022

GENERAL COMMENTS	The authors appropriately answered all the previous comments.
---

REVIEWER	Orso, Daniele University of Udine
REVIEW RETURNED	14-Feb-2022

GENERAL COMMENTS	The authors performed a systematic research to analyze the effect of using general practitioners or nurses on emergency room triage. As far as I have verified the research methodology is solid. The results are plausible and the discussion of these is adequate and complete. The only point that would be discussed more extensively is the repercussion in terms of avoidable mortality or diagnostic error. However, the limit is intrinsic to the available literature, as the authors point out.
---